# ORAI2 Down-Regulation Potentiates SOCE and Decreases Aβ42 Accumulation in Human Neuroglioma Cells

**DOI:** 10.3390/ijms21155288

**Published:** 2020-07-25

**Authors:** Elena Scremin, Mario Agostini, Alessandro Leparulo, Tullio Pozzan, Elisa Greotti, Cristina Fasolato

**Affiliations:** 1Department of Biomedical Sciences, University of Padua, Via U. Bassi 58/B, 35131 Padua, Italy; elena.scremin@studenti.unipd.it (E.S.); agostinimario@icloud.com (M.A.); alessandro.leparulo@unipd.it (A.L.); tullio.pozzan@unipd.it (T.P.); 2Neuroscience Institute—Italian National Research Council (CNR), Via U. Bassi 58/B, 35131 Padua, Italy; 3Venetian Institute of Molecular Medicine (VIMM), Via G. Orus 2B, 35129 Padua, Italy

**Keywords:** Alzheimer’s Disease, amyloid-beta, neuroglioma cells, ORAI2, calcium entry, stores, SOCE

## Abstract

Senile plaques, the hallmarks of Alzheimer’s Disease (AD), are generated by the deposition of amyloid-beta (Aβ), the proteolytic product of amyloid precursor protein (APP), by β and γ-secretase. A large body of evidence points towards a role for Ca^2+^ imbalances in the pathophysiology of both sporadic and familial forms of AD (FAD). A reduction in store-operated Ca^2+^ entry (SOCE) is shared by numerous FAD-linked mutations, and SOCE is involved in Aβ accumulation in different model cells. In neurons, both the role and components of SOCE remain quite obscure, whereas in astrocytes, SOCE controls their Ca^2+^-based excitability and communication to neurons. Glial cells are also directly involved in Aβ production and clearance. Here, we focus on the role of ORAI2, a key SOCE component, in modulating SOCE in the human neuroglioma cell line H4. We show that ORAI2 overexpression reduces both SOCE level and stores Ca^2+^ content, while ORAI2 downregulation significantly increases SOCE amplitude without affecting store Ca^2+^ handling. In Aβ-secreting H4-APPswe cells, SOCE inhibition by BTP2 and SOCE augmentation by ORAI2 downregulation respectively increases and decreases Aβ42 accumulation. Based on these findings, we suggest ORAI2 downregulation as a potential tool to rescue defective SOCE in AD, while preventing plaque formation.

## 1. Introduction

Both the familial (FAD) and sporadic (SAD) forms of AD are characterized by abnormal accumulation of Aβ which leads to the formation of amyloid deposits, the so-called senile plaques, culminating in synaptic dysfunctions, inflammation and neuronal death [1,2,3]. Aβ is the product of APP proteolysis performed by two enzymes belonging to the secretase family. The last proteolytic cleavage, leading to Aβ production, is due to either presenilin-1 (PS1) or presenilin-2 (PS2), whose mutations, together with those of APP, are linked to FAD [4]. The enzyme, comprising either PS1 or PS2, called γ-secretase, produces several proteolytic variants of the Aβ peptide, among which Aβ40 and Aβ42 are the most common isoforms [1,2,3]. In AD patients as well as in AD mouse models, the stoichiometric ratio between the two variants is altered, favouring the longer version, which is more prone to oligomerisation and deposition [3,5]. Further investigations on the pleiotropic roles of presenilins, led to the discovery of their effect on cellular Ca^2+^ handling: specifically, when mutated, they alter the endoplasmic reticulum (ER) Ca^2+^ content as well as the related refilling mechanism, known as SOCE [6,7,8,9,10,11,12,13,14,15,16,17,18,19,20,21,22]. Aβ accumulation was variably suggested to be a Ca^2+^-sensitive event, with SOCE playing a primary role. However, up until now conflicting results have been reported. SOCE inhibition and activation were respectively linked to Aβ augmentation and decrement [7,23,24], whereas other groups reported that SOCE activation induces Aβ42 accumulation [25,26,27].

SOCE is finely tuned by a biochemical mechanism that is comprised of the ER resident Ca^2+^ sensor stromal interaction molecule (STIM1/2), and the plasma membrane (PM) channel-forming subunit ORAI (ORAI1/2/3) [28,29]. Upon depletion of the store Ca^2+^ content, STIM proteins oligomerize, leading to the formation of punctae at the ER membrane, which in turn recruit ORAI subunits at the PM. ORAIs are small 25–30 kDa proteins, characterized by four trans-membrane domains; when contacted by STIMs they oligomerize into hexamers to form channels with high Ca^2+^ selectivity and very low unitary conductance [30] that are responsible of the Ca^2+^ release-activated Ca^2+^ current (I_CRAC_) [31]. Whereas STIM1 and ORAI1 proteins have been deeply investigated [32], the same does not hold true for STIM2, ORAI2 and ORAI3 [28,33]. Only recently has the ORAI2 functionality in the immune system started to be well understood [34,35,36,37,38,39,40]. Conversely, in the nervous system, the role of SOCE role and its components remain obscure [41,42,43]. All three subunits are expressed both in neurons and glial cells [11,37,42,44,45,46,47]. In neurons, ORAI1 appears dispensable for SOCE [46], but it has been implicated in neuronal excitability [45,48], whereas ORAI2 seems to be part of a neuronal SOC (nSOC) based on TRPC6 and activated by STIM2, and its activity is impaired in mouse AD models [46]. In glial cells, SOCE plays a major role in the Ca^2+^-based excitability that characterizes these cells both in culture [14,42,47,49] and in situ [50,51] studies.

In HEK293T cells, ORAI2 was also suggested to be part of the elusive ER Ca^2+^ leak channel [52], therefore offering a possible link between ER and SOCE alterations found in different FAD models ranging from cell lines, fibroblasts and induced pluripotent stem cells from FAD patients [19,20,53] to neurons and astrocytes from AD mouse models [9,10,11,14,46,54].

In this study, we focus on the human neuroglioma cell line H4 and its clone H4-APPswe that stably expresses the FAD-linked APP Swedish mutation. These cells are a good model to modulate glial SOCE components and, at the same time, to verify how this modulation affects Aβ accumulation. Of note, glial cells, among the other relevant features also directly participate in Aβ production [55,56,57,58], especially upon increased cellular stress caused by different environmental factors [59] and neuroinflammation [60].

In neuroglioma cells, we have characterized the effect of increased and decreased ORAI2 expression on SOCE level and ER Ca^2+^ content and found that ORAI2 downregulation significantly increases SOCE amplitude while leading to a marked reduction in the Aβ42/Aβ40 ratio in the extracellular environment. We suggest ORAI2 as a novel therapeutic target in AD because its downregulation allows for the rescue of SOCE reduction, and at the same time, it reduces Aβ42 secretion by glial cells.

## 2. Results

### 2.1. ORAI2 Modulates SOCE in Human Neuroglioma Cells

In this work, we first addressed whether modulation of ORAI2 protein level modifies Ca^2+^ dynamics in the human neuroglioma cell line H4, a widely used cell model for non-neuronal cells of the central nervous system (CNS). In these cells, we carried out experiments with cytosolic aequorin (cytAEQ) to measure Ca^2+^ changes upon expression of the Ca^2+^ probe together with human ORAI2 and STIM1, a condition that allows optimal SOCE activation upon store depletion [30,61]. Upon 24 h from transfection, we evaluated the effect of ORAI2 overexpression on SOCE amplitude by employing the Ca^2+^-addback protocol [62] (Figure 1). A complete store depletion was obtained by challenging the cells with histamine (Hist, 100 μM) and the reversible blocker of the SERCA pump cyclopiazonic acid (CPA, 20 μM) in a Ca^2+^-free mKRB, containing EGTA (0.6 mM). Subsequently, the cells were perfused with a mKRB containing CaCl_2_ (1 mM) and CPA (20 μM) to avoid SOCE inactivation. When compared to cells expressing only STIM1 (Figure 1A, black trace, CTRL), the expression of ORAI2-myc (Figure 1A, cyan trace) reduced both ER Ca^2+^ release (Figure 1B) and SOCE (Figure 1C). Quantitatively, significant reductions were obtained both in terms of peak and area under the curve (AUC) (−34 ± 8% and −35 ± 14%, for Ca^2+^ release and −28 ± 9% and −41 ± 7%, for SOCE, respectively). Similar data were obtained in H4 cells expressing ORAI2-myc also tagged with a monomeric Venus (ORAI2-myc-mVenus) (Figure 1A, red trace and Figure 1B,C).

Altogether, we can state that the overexpression of ORAI2 together with STIM1 decreases both ER Ca^2+^ release and SOCE induced by store depletion. These findings expand previous observations obtained in HEK293T cells, but in the absence of STIM1 overexpression [52]. They also suggest that addition of a C-terminal fluorescent-tag mVenus to ORAI2-myc does not interfere with its functionality, since its expression mimics the Ca^2+^-related phenotype obtained with the expression of ORAI2-myc.

By Western blot and immunofluorescence analyses, we verified that, in H4 cells, the two constructs had similar expression levels (Figure 2A,B) and subcellular localization (Figure 2C,D). Both constructs similarly stained a perinuclear structure, likely the Golgi apparatus (GA), a vesicular network and the plasma membrane (PM).

Based on siRNA perturbation experiments and computational data, it was also suggested that, in HEK293T cells, ORAI2 contributes to ER Ca^2+^ leak [52]. Our data indicate that, in these cells, endogenous ORAI2 localizes primarily in the early endosomes, as estimated by immunofluorescence studies (Appendix A). The Pearson’s coefficient was in fact larger for ORAI2 colocalization with markers of the early endosomes (Appendix A) with respect to a marker of ER (Appendix A) or late endosomes (Appendix A), being respectively: 0.26 ± 0.07 (ER-GFP), 0.28 ± 0.04 (LBPA), 0.44 ± 0.09 (rab5) and 0.61 ± 0.07 (EEA1) (Appendix A). However, only with this latter marker was there a statistically significant difference (*p* < 0.05 compared to ER-GFP and *p* < 0.01 compared to LPBA).

To better define ORAI2 localization in H4 cells, we exploited the overexpression of ORAI2-myc-mVenus with the mCherry protein, targeted to different organelles. In intact H4 cells, ORAI2-myc-mVenus was spread over a large vesicular network (Figure 3), which was however clearly distinct from ER membranes (Figure 3A) and showed only a minor overlap with the endosomal compartment (Figure 3B). ORAI2 better localized at the PM upon live staining with wheat germ agglutinin (WGA) (Figure 3C). Of note, the PM localization of ORAI2-mVenus was clearly visible upon cell permeabilization (Figure 3D, see also Figure 2D).

From a functional point of view, we investigated whether the reduction in ER Ca^2+^ release, observed in ORAI2-overexpressing cells, was the consequence of a decrease in SOCE amplitude or IP_3_R signalling, or even a direct effect of ORAI2 on ER Ca^2+^ handling. We directly evaluated the effect of ORAI2 at the ER Ca^2+^ level by expressing ORAI2 together with G-CEPIA1er, a genetically encoded, ER-targeted Ca^2+^ probe with very low Ca^2+^ affinity and good dynamic range [63]. Surprisingly, G-CEPIA1er expression in H4 cells was extremely variable being also hampered by a high level of instability and photobleaching. We thus decided to take advantage of HeLa cells for this type of experiment because of a much more stable and reliable G-CEPIA1er signal. In particular, with respect to control (CTRL), void-vector transfected cells, ORAI2-myc overexpression decreased the resting ER fluorescence (Figure 4A,B). Since G-CEPIA1er is a non-ratiometric Ca^2+^ indicator, we obtained an indirect estimation of the ER Ca^2+^ content, calculated as the maximal change obtained after CPA and ionomycin addition (Figure 4A). Upon ORAI2-myc overexpression, the maximal ER Ca^2+^ decrease was significantly reduced by about 40% in ORAI2-myc-expressing cells (Figure 4B).

We also estimated the ER Ca^2+^ refilling process in permeabilized HeLa cells expressing G-CEPIA1er (Figure 4C). In permeabilized cells, ORAI2 overexpression significantly impaired the ER Ca^2+^ refilling process by more than 62 ± 7% as estimated by maximal CaCl_2_ (1 mM) addition (Figure 4D). Of note, in ORAI2 overexpressing cells, the rate of Ca^2+^ uptake was markedly different from control cells even at the initial stage of the refilling process (Figure 4C), suggesting an impairment of the SERCA pump activity rather than an effect on ER Ca^2+^ leakage. This latter should in fact become apparent only after a certain level of store replenishment. Altogether these findings support the idea that upregulation of ORAI2 impairs the SERCA activity independently of its effect on SOCE level.

### 2.2. SOCE Levels and Aβ-Secretion in Neuroglioma Cells

Different groups have suggested a close relationship between cytosolic Ca^2+^ levels and Aβ accumulation, however with contrasting results [7,24,25,27]. To investigate this issue, we employed the H4-APPswe cell line, that stably express the human APP Swedish mutation (K670/M671L) and accumulates Aβ in the culture medium [64].

By exploiting a pharmacological approach in Aβ-secreting cells, we identified an inverse relationship between SOCE and Aβ accumulation. When H4-APPswe cells were incubated for 24 h in the culture medium with CPA (0.5 μM) or BTP2 (1 μM) to activate or inhibit SOCE, respectively, the Aβ42 that accumulates in the culture medium was either decreased by CPA (−11 ± 1%, *p* < 0.01), or increased by BTP2 (+11 ± 4%, *p* < 0.05), if compared to vehicle (DMSO 0.1%) treated cells (Appendix A).

It was demonstrated that ORAI2 knockout increases SOCE in mouse T lymphocytes [35], we thus hypothesized that augmenting SOCE by downregulating ORAI2 would reduce Aβ accumulation by H4-APPswe cells. We firstly checked how ORAI2 modulates SOCE in this cell line. To detect SOCE we thus employed the Ca^2+^-addback protocol we adopted for H4 cells, as described in Figure 1. Unfortunately, very heterogeneous Ca^2+^ increases were observed following CaCl_2_ addition to H4-APPswe cells upon store depletion with histamine and CPA in Ca^2+^-free, EGTA-containing mKRB (Appendix A). Large Ca^2+^ rises were observed also in the absence of store depletion that were insensitive to a two-minute-treatment with different Ca^2+^ channel inhibitors, such as GdCl_3_ (1 μM), nimodipine (Nim, 1 μM) plus verapamil (Ver, 10 μM), or carbenexolone (CBX, 50 μM) (Appendix A). These Ca^2+^ rises were likely due to Ca^2+^ entry through other channels, possibly activated by removal of extracellular Ca^2+^. To overcome this problem, cells were challenged with CPA (20 μM) in mKRB in the continuous presence of extracellular CaCl_2_ (1 mM) (Appendix A). The contribution of Ca^2+^ release to Ca^2+^ rises was evaluated in parallel experiments by adding CPA in a Ca^2+^-free, EGTA-containing, mKRB (Appendix A). Under these conditions, the Ca^2+^ release, induced by CPA, was rather modest and returned to the resting Ca^2+^ level in a couple of minutes. Furthermore, given the strong dependence of SOCE on membrane hyperpolarization [31], to reduce the variability due to membrane potential, additional experiments were carried out in KCl-mKRB, upon substitution of extracellular NaCl with equimolar KCl (100 mM). In this condition, to compensate for the reduced driving force for Ca^2+^ entry, the CaCl_2_ concentration was increased from 1 to 10 mM [62]. Similarly to experiments in H4 cells, STIM1 was expressed in the absence (CTRL) or presence of ORAI2. It is worth mentioning that, when store depletion occurs in a Ca^2+^-containing medium, the initial rate of Ca^2+^ entry following CPA addition is not indicative solely of SOCE kinetics, since the signal is contaminated by the Ca^2+^ rise due to Ca^2+^ release from intracellular stores. We thus estimated differences in SOCE amplitude by Ca^2+^ peak and AUC, measured at two minutes, when Ca^2+^ release was practically exhausted (Appendix A).

In H4-APPswe cells, bathed in Ca^2+^-containing mKRB, overexpression of ORAI2-myc, together with STIM1, significantly decreased SOCE by about 50%, in terms of both peak and AUC (Appendix A); similar results were obtained when using KCl-mKRB (Appendix A). Thus, as shown in H4 cells, SOCE reached lower levels than in control cells upon ORAI2 upregulation. Conversely, ORAI2 downregulation by siRNA (siORAI2) increased SOCE by about 80%, when measured in mKRB (Figure 5A,B) and 70% in KCl-mKRB (Figure 5C,D), both in terms of peak amplitude and AUC. As expected, ORAI1 downregulation reduced SOCE by more than 60%, in both standard- and KCl-mKRB (Appendix A).

To investigate the effect of ORAI2 (plus STIM1) overexpression on the store Ca^2+^ content of H4-APPswe cells, we induced maximal Ca^2+^ release by histamine and CPA in a Ca^2+^-free, EGTA-containing medium. Similarly, to what observed in H4 cells (Figure 1A,B), ORAI2 overexpression significantly reduced ER Ca^2+^ release by 32 ± 11% in terms of area (Figure 6A,B). Instead, upon ORAI2 downregulation by siRNA, no difference was found in Ca^2+^ released from the IP_3_-sensitive stores (Figure 6C,D).

We also investigated whether ORAI2 overexpression modifies the basal Ca^2+^ level of H4-APPswe cells. While no difference was found in cells bathed in mKRB containing CaCl_2_ (1 mM) (CTRL = 100 ± 7%, *n* = 26; ORAI2 = 112 ± 6%, *n* = 31, *p* = 0.11), ORAI2 overexpression almost doubled the resting Ca^2+^ level of cells bathed in KCl-mKRB, containing high CaCl_2_ (10 mM) (CTRL = 100 ± 13%, *n* = 31; ORAI2 = 176 ± 14.0%, *n* = 32, *p* < 0.001). However, due to the intrinsic low sensitivity of cytAEQ in reporting resting Ca^2+^ levels, this issue was also investigated with nuclear-targeted GCaMP6 (H2B-GCaMP6), a probe with higher Ca^2+^ affinity with respect to cytAEQ and a very large dynamic range (K_d_ = 375 nM) [65], whose signal is synchronized with the cytosolic one [66]. In mKRB containing CaCl_2_ (1 mM), ORAI2 overexpression caused a large increase in basal Ca^2+^ levels (Appendix A), as estimated by the ratio change occurring at rest upon addition of EGTA (*ΔR* = R_Ca_−R_EGTA_), being equal to 0.7 ± 0.2 and 12.4 ± 1.7 for CTRL and ORAI2 expressing cells, respectively (*p* < 0.001). In contrast, there was no significant difference in cells upon ORAI2 downregulation (Appendix A, *ΔR* = 0.23 ± 0.04 and 0.55 ± 0.16 in siCTRL- and siORAI2-transfected cells, respectively).

In summary, ORAI2 overexpression modifies SOCE, as well as the store Ca^2+^ content and resting cytosolic levels. In contrast, ORAI2 downregulation selectively increases SOCE without altering the store Ca^2+^ content or basal Ca^2+^ levels. Since our data show that the SOCE inhibitor BTP2 increases Aβ42 accumulation (Appendix A), it is expected that upregulation of SOCE will negatively affect this parameter. ELISA experiments were performed on conditioned media of H4-APPswe cells collected at 48 h upon transfection with siORAI2 or control siRNA. Under these conditions, the Aβ42 level was reduced by 44 ± 10%, while the Aβ40 level was increased by 86 ± 13%, resulting in an approximately 70% decrease of the Aβ42/Aβ40 ratio (Figure 7).

## 3. Discussion

According to recent models, the store-operated Ca^2+^ entry (SOCE) based on I_CRAC_ is due to Ca^2+^ permeation across hexameric channels formed by ORAI (1/2/3) subunits [28,30]. While the role of ORAI1 has widely been clarified, the contribution of the other subunits to I_CRAC_ and SOCE emerged only later [28,33]. Thanks to in-depth studies on the immune system, it has elegantly been demonstrated that, in mouse T cells, ORAI2 forms heteromeric channels with ORAI1 and contributes to a reduced Ca^2+^ permeation through I_CRAC_ [35,39]. A negative role for ORAI2 on Ca^2+^ entry was already postulated following overexpression studies in HEK293 cells [67]. In addition, Meyer and co-workers demonstrated that, in HEK293T cells, overexpression of ORAI2 decreases Ca^2+^ release and content, whereas its downregulation exerts an opposite effect [52].

In this work, we addressed two aspects of ORAI2 modulation of Ca^2+^ homeostasis: (i) by using three types of model cells, we investigated the effects of ORAI2 levels (overexpression or downregulation) on the Ca^2+^ content of intracellular stores and the modulation of SOCE; (ii) in the neuroglioma cell line H4-APPswe, we also investigated the effects of SOCE modulation on Aβ production.

Concerning the first aspect, we here provided evidence that modulation of ORAI2 levels tunes the amplitude of SOCE, with an inverse relationship between ORAI2 expression and Ca^2+^ entry induced by store depletion. Furthermore, by using different approaches in intact and permeabilized cells, we demonstrated that ORAI2 overexpression decreases the store Ca^2+^ content, independently of SOCE reduction or IP_3_-signalling. These findings are in part consistent with data previously reported in HEK-293T cells [52]. However, the co-expression of ORAI2 with an excess of STIM1, as reported in this study, allows us to exclude a dominant negative effect of ORAI2 on endogenous ORA1/STIM1 subunits. At high levels, ORAI2 could in fact sequester a substantial amount of STIM1, preventing effective STIM1-ORAI1 interactions at the PM [30,61].

It should be noted that, at variance with HEK-293T cells [52], in neuroglioma cells, ORAI2 downregulation by siRNA leaves unchanged the release of Ca^2+^ from intracellular stores. This last result is not consistent with the hypothesis that endogenous ORAI2 controls the store Ca^2+^ content by modulating the ER Ca^2+^ leak.

In order to try to solve this discrepancy we investigated the subcellular distribution of overexpressed ORAI2 using either a version containing only a myc-tag or one with mVenus-tag fused at the C-terminal. No difference between the two constructs was observed. The overexpression of ORAI2 in H4 cells resulted in the accumulation of the tagged proteins primarily at the PM level and early endosomes, with no significant overlapping with ER marker. Thus while in H4 and HeLa the overexpression of ORAI2 produced Ca^2+^ effects similar to those reported in HEK293T cells, i.e., it reduced the store Ca^2+^ content and increased the resting Ca^2+^ level [52], downregulation of the protein left both parameters unchanged. At the moment, the reason for these discrepancies remains unexplained. Yet, the reduction in store Ca^2+^ level and the slowness in Ca^2+^ re-uptake, observed in permeabilized cells upon ORAI2 overexpression, together with the lack of ORAI2 colocalization with ER markers suggest that the effect of ORAI2 is likely indirect and not consistent with ORAI2 as an ER leak channel. Furthermore, upon ORAI2 downregulation, the lack of effect on the basal Ca^2+^ level excludes a direct role of endogenous ORAI2 in controlling the resting Ca^2+^ concentration, whereas the increase in resting cytosolic Ca^2+^ level, observed upon overexpression, is consistent with the capability of this subunit to form homomeric functional PM Ca^2+^ channels [35,67], as well as with SOCE activation by chronic reduction in the store Ca^2+^ content, as originally suggested [52].

SOCE dysregulation has been widely reported among the complex defects that characterize Ca^2+^ dyshomeostasis in AD [6,7,8,10,11,12,13,14,15,16,17,18,19,20,21,22,46,68,69]. In particular, whereas the role of FAD-linked PS1 and PS2 mutations at the store level was debated [18], the large majority of data converge towards SOCE downregulation, when studied either in cell lines expressing the PS1/2 mutants or in fibroblasts from FAD patients [6,9,10,11,12,13,16,19,20,21,46] as well as in SAD [10,70]. However, SOCE upregulation was also reported in 3xTg AD mice [14].

In neurons, the identification of both SOCE components and role is still controversial [43], while SOCE seems to play a primary role in modulating the Ca^2+^-based excitability in astrocytes [47,49,51,71]. It is worth noting that astrocytes are also directly involved in Aβ production [26,55,56,57,58], while alterations in Ca^2+^-based excitability has been reported in astrocytes from FAD mouse models [54,72,73].

Over the last two decades, different groups have investigated the relationship between SOCE and Aβ production, with different and often divergent results [7,23,24,25,26,27]. In this work, we took advantage of ORAI2 downregulation to study the relationship between SOCE amplitude and Aβ accumulation while avoiding alterations at the store Ca^2+^ level. When this relationship was studied by employing SERCA pump inhibitors [25,26,27] or SERCA2b-siRNA [23] in the presence of extracellular Ca^2+^, a direct relationship between SOCE activation and Aβ42 accumulation was observed. Yet, it was also reported that pharmacological inhibition of SOCE increases Aβ42 accumulation [7]. Furthermore, SOCE activation by overexpression of ORAI1 or STIM1-D76A, a constitutively active form of STIM1, reduces Aβ42 production [24]. We here show that, in Aβ-secreting neuroglioma cells, overnight incubation with CPA or BTP2 (to activate or inhibit SOCE respectively) decreases and increases Aβ42 accumulation, being consistent with an inverse relationship between the two pathways [7,23,24].

The conflicting results obtained with SERCA pump inhibitors by different groups could have different explanations: (i) the use of different cell models; (ii) the different contribution to Ca^2+^ rises played by Ca^2+^ release and Ca^2+^ entry in those cells and (iii) the different roles played by Ca^2+^ on amyloid processing, Aβ production and secretion.

ORAI2 downregulation increases SOCE amplitude without modifying the store Ca^2+^ content thus it permits to test how an increased Ca^2+^ entry through SOCE influences Aβ secretion.

In H4-APPswe cells, downregulating ORAI2 by siRNA decreases Aβ42 but increases Aβ40 accumulation, significantly reducing the Aβ42/Aβ40 ratio. Altogether, these findings suggest that increased Ca^2+^ entry through ORAI channels favours the accumulation of the less amyloidogenic peptide. It is worth noting that the APP Swedish mutation, by itself, does not change the Aβ42/Aβ40 ratio but simply increases the number of secreted peptides, because of its enhancement of β-secretase cleavage. For this reason, we can consider H4-APPswe cells an acceptable model of Aβ accumulation in SAD.

It is known that γ-secretase exists in a dynamic equilibrium of conformational states, with the “closed” conformation being associated with the shift of the enzyme cleavage towards the production of longer, neurotoxic Aβ species; of note, the shift to the pathogenic closed conformation is regulated by Ca^2+^ [74,75]. In neurons, the Aβ42/Aβ40 ratio is strictly dependent on the pattern of spiking activity [74]. In particular, an increase in burst activity, increases cytosolic Ca^2+^ and shifts PS1 towards the open conformation thus augmenting the Aβ40 production and reducing the Aβ42/Aβ40 ratio [74]. In this work, we provide evidence that increasing Ca^2+^ levels by SOCE potentiation similarly favours Aβ40 with respect to Aβ42 secretion; a model explaining the possible linkage between SOCE potentiation, PS conformation, and Aβ production is presented in Figure 8.

A sustained increased in Ca^2+^ entry might also affect the endocytic pathway that controls PS maturation and APP processing through γ-secretase. It is worth noting that, in addition to its PM localization, ORAI2 is widely distributed in a vesicular network, partially overlapping with the endocytic pathway, thus suggesting a possible role of this protein in the regulation of APP processing at the endosomal level. Here, we demonstrated that ORAI2-myc and ORAI2-myc-mVenus are indistinguishable with respect to Ca^2+^ handling by neuroglioma cells, thus providing a useful tool to investigate the relationship between APP processing and SOCE in intact cells.

In conclusion, here, we showed that, in Aβ-secreting neuroglioma cells, downregulation of ORAI2 potentiates SOCE without altering the store Ca^2+^ content and decreases the Aβ42/Aβ40 ratio thus being an interesting tool to restore defective Ca^2+^ entry associated to AD [9,10,54,76], while preventing amyloid seeding, given the direct correlation of this latter with the Aβ42/Aβ40 ratio. Altogether, our data support the idea that SOCE, and particularly ORAI2, could be a potential therapeutic target in AD.

## 4. Materials and Methods

### 4.1. Cell Culture and Transfection

H4, H4-APPswe, HEK293T, and HeLa cells were grown in DMEM (Dulbecco’s Modified Eagle Medium, Merck Life Science, Milan, Italy) containing 10% FCS (fetal calf serum, Merck Life Science, Milan, Italy), supplemented with l-glutamine (2 mM), penicillin (100 U/mL), and streptomycin (100 μg/mL), in a humidified atmosphere containing 5% CO_2_. For cytAEQ and immunofluorescence experiments, cells were seeded onto 13 mm glass coverslips and cultured into 24-well plates. For fluorescent probe experiments, cells were seeded onto 24 mm glass coverslips and cultured into 6-well plates. For Western blot and Enzyme-Linked ImmunoSorbent Assay (ELISA) experiments, cells were seeded directly into 6-well plates. Transfection was performed at 60–70% confluence with the cDNA coding for the proteins of interest or the void-vector (pcDNA3.1) and the cDNA coding for the Ca^2+^ probe in a 4:1 ratio or rab5-mCherry (Addgene #49201) or ER-mCherry (Addgene Plasmid #55041) in a 1:1ratio using TransIT LT1 Transfection Reagent (Mirus, MIR 2305) (3 μL/μg DNA). For Wheat Germ Agglutinin (WGA, Texas Red^®^-X Conjugate, Thermo Fisher scientific, Monza, Italy, #W21405) staining, cells were incubated for 10 min at 37 °C in a Ca^2+^-containing mKRB supplemented with WGA (5 μM), then washed in PBS and fixed. For RNAi experiments, cells were transfected using Lipofectamine RNAi-MAX transfection reagent in complete culture medium (Thermo Fisher scientific, Monza, Italy, #13778150) (3 μL/mL). siRNAs (human Orai2 MISSION esiRNA EHU060251, human Orai1 MISSION esiRNA EHU120081, Universal Negative Control #1 MISSION siRNA SIC001) were used at a final concentration of 25 nM. Western blots and immunofluorescence were usually performed 24 h after transfection, while Ca^2+^ measurements were performed 24 and 48 h after transfection, for overexpression and downregulation, respectively.

### 4.2. Aequorin Measurements

Upon transfection, the cells were incubated for 1 h with coelenterazine (5 μM) in modified Krebs-Ringer Buffer (mKRB, composition in mM: 140 NaCl, 2.8 KCl, 2 MgCl_2_, 10 4-[2-hydroxyethyl]-1-piperazineethanesulfonic acid – HEPES -, 10 glucose, pH 7.4 at 37 °C) containing CaCl_2_ (1 mM) and then transferred to the perfusion chamber. The cytAEQ measurements were carried out in mKRB containing CaCl_2_ (1 mM) at 37 °C. Where indicated, in mKRB, NaCl (100 mM) was substituted by KCl (100 mM) (KCl-mKRB composition in mM: 43 NaCl, 100 KCl, 2 MgCl2, 10 HEPES, 10 glucose, pH 7.4 at 37 °C) to set to zero the membrane potential. Agonists and other drugs were added to the same medium, as specified in the figures. The experiments were terminated by lysing the cells with digitonin (100 μM) in a hypotonic Ca^2+^-rich solution (10 mM CaCl_2_) in H_2_O to discharge the remaining unused AEQ pool. Signal detection and analysis was carried out as previously described [77].

### 4.3. Ca^2+^ Imaging

HeLa cells expressing G-CEPIA1er fluorescent probe and H4-APPswe cells expressing H2B-GCaMP6 fluorescent probe were bathed in mKRB containing CaCl_2_ (1 mM) at 37 °C. When indicated, cells were bathed in a Ca^2+^-free mKRB containing EGTA (0.6 mM) or in a Ca^2+^-free intracellular solution also containing EGTA (0.05 mM) of the following composition in mM (130 KCl, 10 NaCl, 1 MgCl_2_, 2 succinic acid and 20 HEPES, pH 7.0 at 37 °C). Cells were analyzed using an inverted microscope (Zeiss Axiovert 100, Jena, Germany) with a Fluar 40X oil objective (NA 1.30). Excitation light (480 nm for G-CEPIA1er, 410/10 nm and 475/10 nm for H2B-GCaMP6) was produced by a monochromator (Polychrome V; TILL Photonics, Graefelting, Germany) and passed through a dichroic mirror DRLP (505 ext XF73). The emitted fluorescence was collected by a bandpass filter (500–530 nm). Images were acquired using a cooled CCD camera (SensiCam QE, PCO, Kelheim, Germany). All filters and dichroic mirrors were from Chroma Technologies (Bellow Falls, VT, USA). Images were collected at 0.2 Hz with 200 ms exposure time. Cells were mounted into an open-topped chamber and maintained in mKRB medium. Additions were made in the same or different medium, as specified in the figures. G-CEPIA1er experiments: data are presented as ΔF / F_0_ or ΔF / F_min_ values (where ΔF are fluorescence changes at any time, F_min_ is the fluorescence value upon ionomycin addition and F_0_ the baseline value). H2B-GCaMP6 experiments: data are presented as ΔR values (where ΔR is the change of the F_470_ / F_410_ emission intensity ratio at any time.

### 4.4. Materials

CPA and ionomycin were from Calbiochem (Merck Life Science, Milan, Italy). All other materials were analytical of the highest available grade were purchased from Sigma-Aldrich (Merck Life Science, Milan, Italy).

### 4.5. Statistical Analyses

Data were analyzed using Origin 8.0 SR6 (OriginLab Corporation, Northampton, MA, USA) and Microsoft Excel 2010 (Microsoft Corporation, Redmond, WA, USA). All data are representative of at least 3 independent experiments. Numerical values presented throughout the text are mean ± S.E.M. (*n* = number of independent experiments or cells, as indicated). Significance was evaluated by the non-parametric Kruskal–Wallis test. Where the Kruskal–Wallis test resulted in the existence of a pair of different populations, differences between means were tested with the Wilcoxon–Mann–Whitney test. Boxplots show median, mean, the 25th and 75th percentile of the distributions, while whiskers indicate the upper and lower extremes. * *p* < 0.05, ** *p* < 0.01, *** *p* < 0.001.

## Figures and Tables

**Figure 1 ijms-21-05288-f001:**
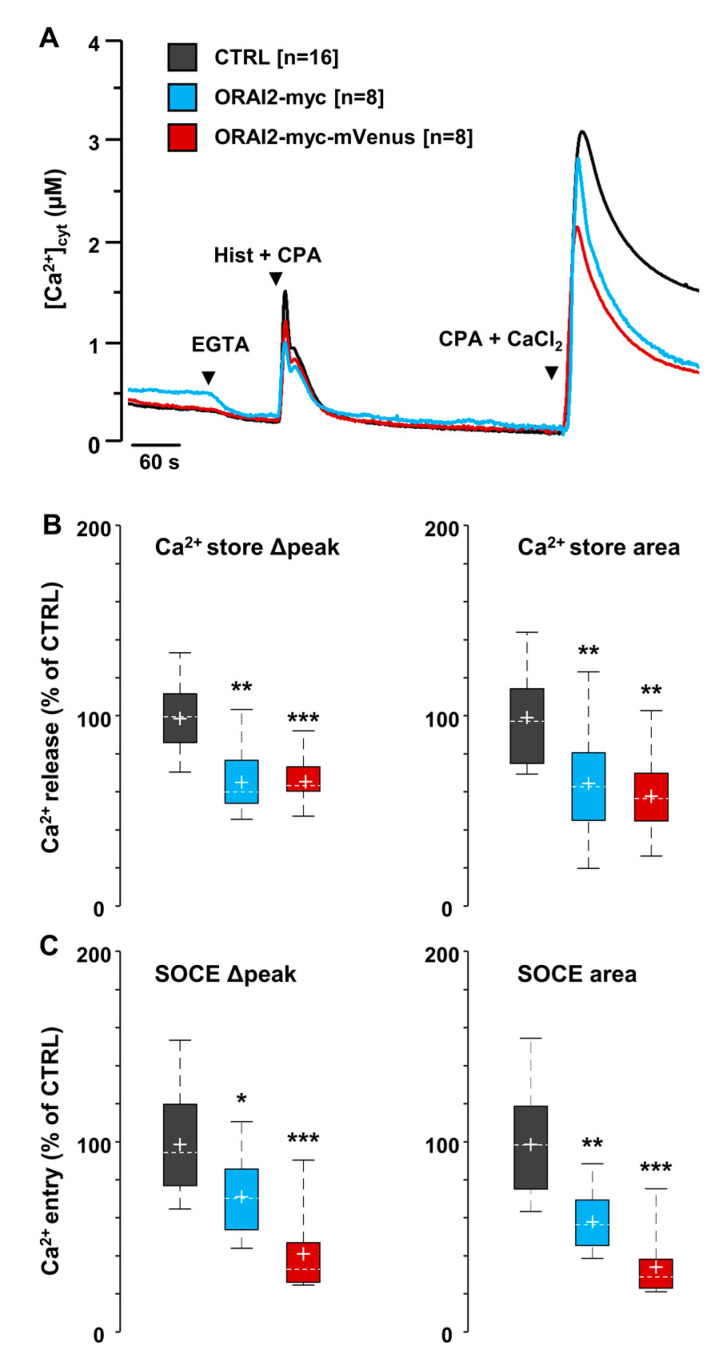
ORAI2 overexpression reduces ER Ca^2+^ release and SOCE in H4 cells. Representative traces of H4 cells expressing ORAI2-myc or ORAI2-myc-mVenus, STIM1 and cyt-AEQ in a 2:6:1 ratio. (**A**) Cells were bathed, under perfusion at 37 °C, in mKRB containing CaCl_2_ (1 mM). Upon shifting to a Ca^2+^-free mKRB containing EGTA (0.6 mM), cells were challenged with histamine (Hist, 100 μM) and CPA (20 μM) in the same medium. After 8 min, cells were bathed in mKRB containing CaCl_2_ and CPA to detect maximal SOCE amplitude. Boxplots show Ca^2+^ release (**B**) and SOCE **(C**) peak height and AUC, upon baseline subtraction. Data are expressed as percentage of control (CTRL) cells expressing only STIM1 and cyt-AEQ. Coverslips were from at least three independent cell batches. Wilcoxon-Mann-Whitney test (* *p* < 0.05; ** *p* < 0.01; *** *p* < 0.001).

**Figure 2 ijms-21-05288-f002:**
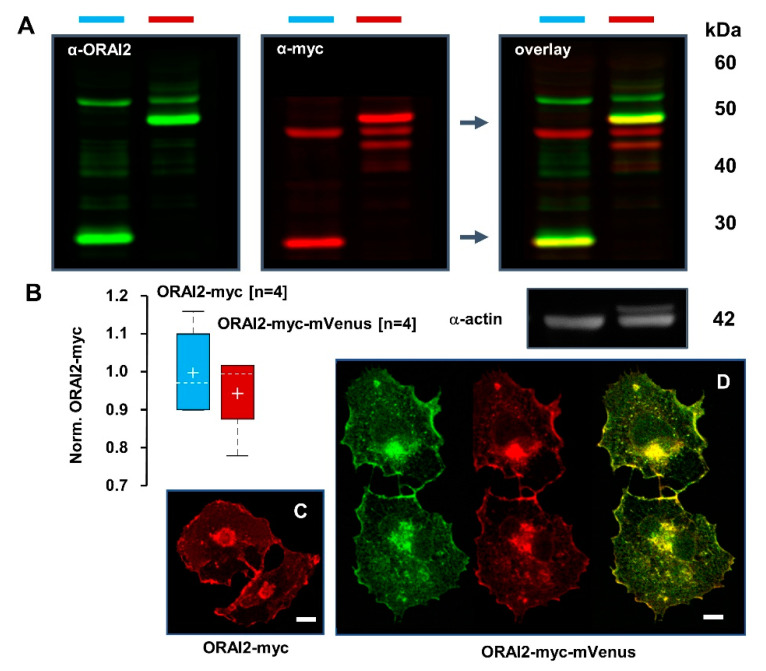
ORAI2-myc and ORAI2-myc-mVenus were similarly expressed in H4 cells. (**A**) Western blots from H4 cells expressing ORAI2-myc or ORAI2-myc-mVenus at 24 h, showing α-ORAI2 (left), α-myc (middle) and their overlay (right). Yellow color indicates the specific Orai2-myc and Orai2-myc-mVenus bands. (**B**) Densitometric analysis of ORAI2-myc and ORAI2-myc-mVenus bands at 25 and 50 kDa upon normalization to α-actin (Norm. ORAI2-myc). (**C**, **D**) Confocal images of ORAI2 localization in H4 cells upon overexpression of ORAI2-myc (**C**) or ORAI2-myc-mVenus (**D**). mVenus fluorescence is shown in green (left), myc-staining in red (middle) and the overlay in yellow (right); scale bars 10 μm.

**Figure 3 ijms-21-05288-f003:**
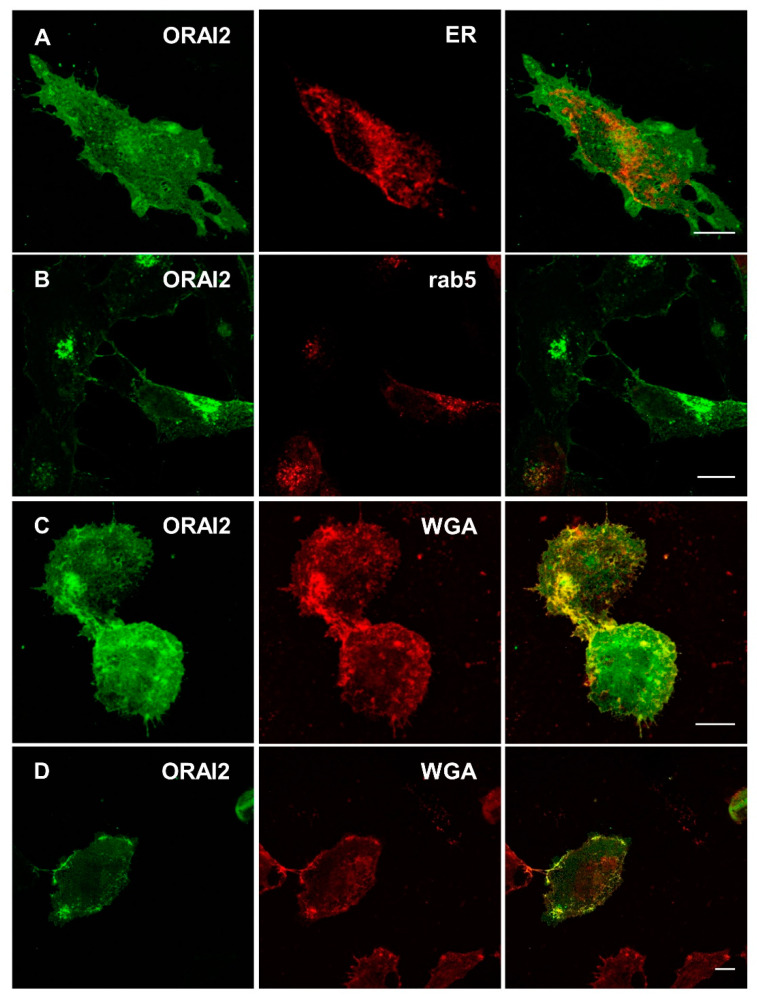
Localization of ORAI2-myc-mVenus in H4 cells. Confocal images of intact H4 cells 24 h upon transfection with the cDNA coding for ORAI2-myc-mVenus (green, left) together with ER- (**A**) or rab5-mCherry (**B**) (red, middle) and their overlay (right). The cells expressing ORAI2-myc-mVenus were also stained with Wheat Germ Agglutinin (WGA) either before (**C**) or after cell permeabilization (**D**); scale bars 10 μm.

**Figure 4 ijms-21-05288-f004:**
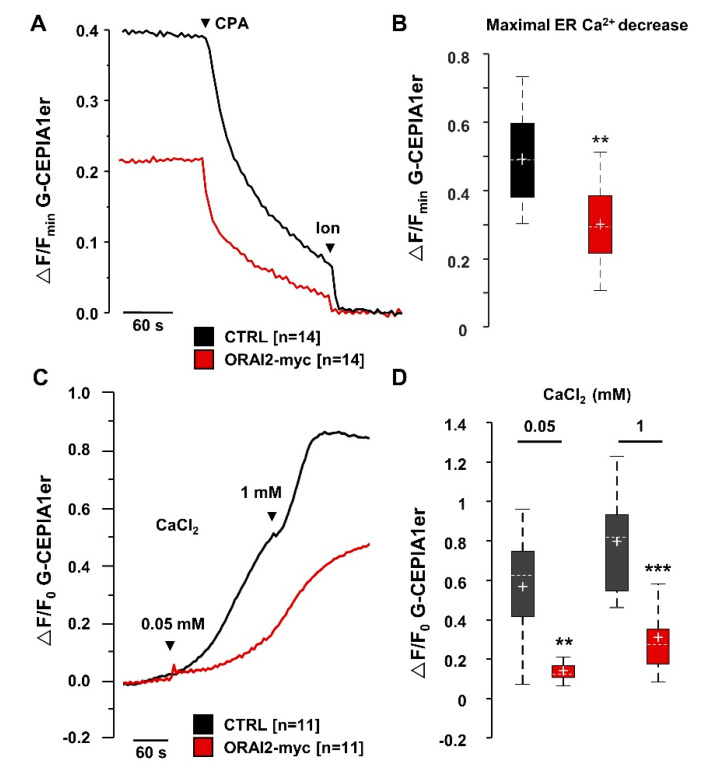
ORAI2-myc overexpression decreases the ER Ca^2+^ content and refilling process. HeLa cells were co-transfected with the cDNAs coding for ORAI2-myc or with the void-vector (CTRL) and G-CEPIA1er in a 4:1 ratio. (**A,B**) Cells were bathed for two minutes in Ca^2+^-free mKRB, containing EGTA (0.6 mM), then challenged with CPA (20 μM) and ionomycin (Ion, 1 μM), where indicated by the arrows. (**A**) Representative traces of G-CEPIA1er fluorescence (F) normalized to the value measured after ionomycin addition (F_min_) and expressed as (ΔF/F_min_). (**B**) Boxplots of the maximal ER-Ca^2+^ decrease (ΔF/F_min_) obtained upon CPA and ionomycin addition, an indication of the ER Ca^2+^ content (*n* = 14, number of coverslips for each condition, from three independent cell batches, Wilcoxon-Mann-Whitney test ** *p* < 0.01). (**C,D**) Cells were pre-depleted with CPA (20 μM) in Ca^2+^-free mKRB, containing EGTA (0.6 mM). Upon CPA washing, cells were exposed to an intracellular solution containing EGTA (50 μM). Cells were permeabilized for 20 s with digitonin (100 μM) and store refilling was induced by CaCl_2_ addition in the presence of ATP (100 μM). (**C**) Representative traces of the maximal fluorescence changes (ΔF) normalized to the initial basal level (F_0_). (**D**) Boxplots of ΔF/F_0_ measured upon CaCl_2_ addition, a rough estimate of the ER Ca^2+^ concentration (*n* = 11, number of coverslips from three independent cell batches, Wilcoxon-Mann-Whitney test, ** *p* < 0.01; *** *p* < 0.001).

**Figure 5 ijms-21-05288-f005:**
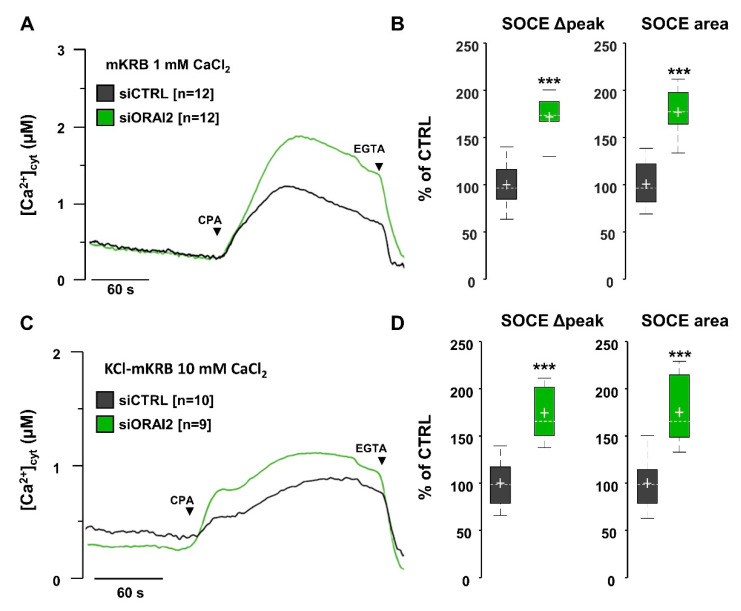
ORAI2 downregulation increases SOCE in H4-APPswe cells. Cells were co-transfected with siCTRL or siORAI2 (25 nM) plus the cDNA coding for cytAEQ. After 48 h, cells were bathed at 37 °C in mKRB (**A,B**) or KCl-mKRB (**C,D**) containing 1 or 10 mM CaCl_2_, respectively. Cells were then treated with CPA (20 μM) to activate SOCE. At the end of the experiment, cells were bathed in Ca^2+^-free, EGTA-containing medium to stop Ca^2+^ entry. (**A,C**) Representative traces are shown for siCTRL (black) and siORAI2 (green) transfected cells, respectively. (**B,D**) Boxplots of SOCE Δpeak and AUC, measured within two minutes upon baseline subtraction. Data are expressed as percentage of CTRL, *n* = number of coverslips, Wilcoxon-Mann-Whitney test *** *p* < 0.001.

**Figure 6 ijms-21-05288-f006:**
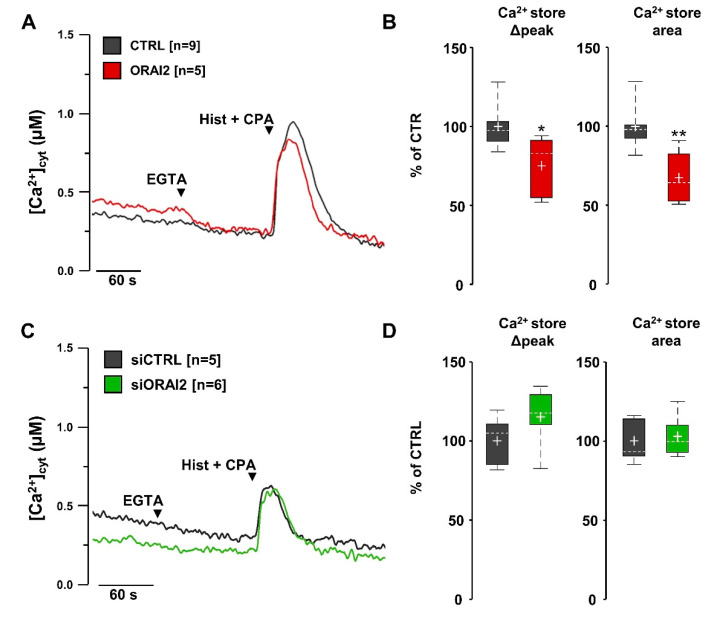
Overexpression, but not downregulation, of ORAI2 modulates the IP_3_-sensitive Ca^2+^ stores of H4-APPswe cells. In H4-APPswe cells, overexpression of ORAI2-myc (plus STIM1) decreases the IP_3_-induced ER Ca^2+^ release while its downregulation by siRNA has no effect. Cells were co-transfected with cyt-AEQ together with void-vector (CTRL) or cDNAs coding for ORAI2 and STIM1 (**A,B**) or siRNAs (siCTRL or siORAI2) (**C,D**). Cells were bathed for 2 min at 37 °C with mKRB containing CaCl_2_ (1 mM), then for one minute with a Ca^2+^-free mKRB, containing EGTA (0.6 mM), before being challenged with histamine (Hist, 100 μM) and CPA (20 μM) in the same medium, as indicated by the arrows. (**A,C**) Representative traces for CTRL (black), ORAI2-overexpressing (red) or siORAI2-transfected (green) cells, respectively. (**B,D**) Boxplots of Ca^2+^ release, in terms of peak amplitude and AUC, upon baseline subtraction. Data are expressed as percentage of CTRL, *n* = number of coverslips, Wilcoxon-Mann-Whitney test, * *p* < 0.05, ** *p* < 0.01.

**Figure 7 ijms-21-05288-f007:**
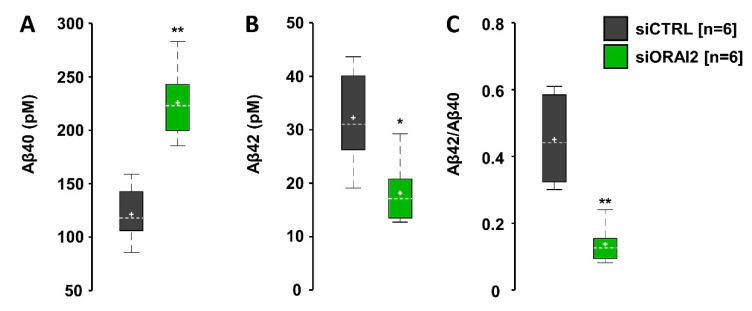
ORAI2 downregulation reduces Aβ42/Aβ40 ratio. H4-APPswe cells were transfected with ORAI2-siRNA (siORAI2) or control-siRNA (siCTRL). Boxplots of Aβ40 (**A**), Aβ42 (**B**) levels and their ratio (**C**), detected in the conditioned media at 48 h (*n* = number of samples, Wilcoxon-Mann-Whitney test, * *p* < 0.05; ** *p* < 0.01).

**Figure 8 ijms-21-05288-f008:**
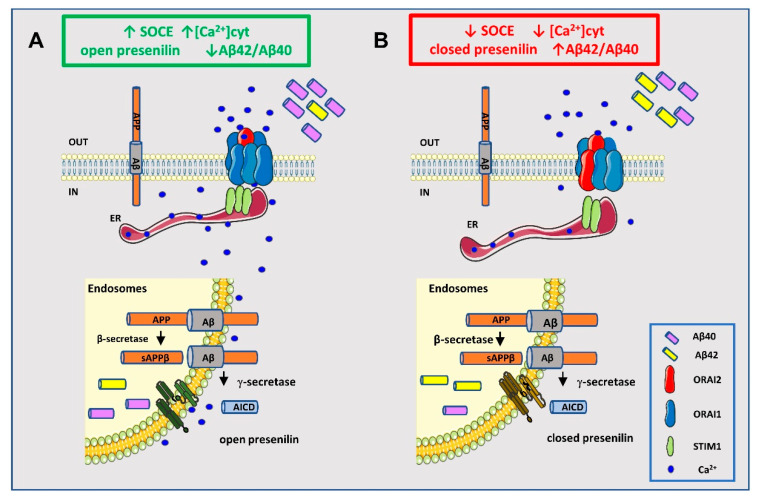
Hypothetical model of the crosstalk between SOCE and Aβ production. (**A**) Higher SOCE levels by ORAI2 downregulation increase the cytosolic Ca^2+^ concentration, favouring the open conformation of presenilin and the accumulation of the shorter peptide Aβ40 (74,75).(**B**) Lower SOCE levels, caused by inhibitors or by reduced STIM1 availability, as observed in AD (12,13,70), favour the closed conformation of presenilin with the accumulation of the longer peptide Aβ42; AICD (APP intracellular domain).

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
