# Peer review of "ORAI2 Down-Regulation Potentiates SOCE and Decreases Aβ42 Accumulation in Human Neuroglioma Cells"

_ijms, 2020, doi:10.3390/ijms21155288_

Round 1

Reviewer 1 Report

Summary

The manuscript by Scremin et al. presents a new finding related to the expression levels of ORAI2 and its impact on the SOCE and Ca+2 levels in the AB-secreting neuroglioma cells and evaluates a potential role of ORAI2 in the context of FAD treatment. While this study mainly reports in vitro results (coming from HEK293T cell lines, etc.), the findings reported in this article address an outstanding issue of overexpression or downregulation of ORAI2 phenotypical aspects that could play a key role in FAD. Overall, this study is performed thoroughly while considering the intricate elements of evaluating the effects of ORAI2 expression levels. It would help researchers working in the field of AD find a better way of designing new drugs targeting the downregulation of ORAI2. As the findings reported in this study stems from results obtained by in vitro experiments, a further validation coming from in vivo (for example, FAD related mice models) would be vital confirming the beneficial role of ORAI2 as a potential target. The finding reported here is essential and is certainly of interest to the researchers working in the AD field. There are, however, some minor changes that could help elevate the message of this study.  

Minor Points

While the authors discuss different aspects of FAD-linked APP Swedish mutation, a more detailed discussion on the type and location (i.e., domain) of the mutation, and how it might impact interactions at the molecular level would help capture the message. A graphical illustration should be sufficient, also depicting the event cascade through the Aβ accumulation.

Similarly, a graphic depicting the role of ORAI2 and the impact on the levels of SOCE in AD is essential and should be added, perhaps in the discussion section.

Author Response

Dear Editor,

We thank you the Reviewers for the precious comments. Here are the detailed answers to the specific points. The revised version shows the relevant changes in yellow. Few typos have also been corrected.

Reviewer 1

Summary

The manuscript by Scremin et al. presents a new finding related to the expression levels of ORAI2 and its impact on the SOCE and Ca+2 levels in the AB-secreting neuroglioma cells and evaluates a potential role of ORAI2 in the context of FAD treatment. While this study mainly reports in vitro results (coming from HEK293T cell lines, etc.), the findings reported in this article address an outstanding issue of overexpression or downregulation of ORAI2 phenotypical aspects that could play a key role in FAD. Overall, this study is performed thoroughly while considering the intricate elements of evaluating the effects of ORAI2 expression levels. It would help researchers working in the field of AD find a better way of designing new drugs targeting the downregulation of ORAI2. As the findings reported in this study stems from results obtained by in vitro experiments, a further validation coming from in vivo (for example, FAD related mice models) would be vital confirming the beneficial role of ORAI2 as a potential target. The finding reported here is essential and is certainly of interest to the researchers working in the AD field. There are, however, some minor changes that could help elevate the message of this study.  

We thank the referee for the detailed analysis of the manuscript and the appreciation of our work. We absolutely agree with the Reviewer about the importance to validate the results obtained in vitro using in vivo models. Indeed, this is our next goal and we have just started to address ORAI2 downregulation by a miRNA approach under a GFAP promoter. Unfortunately our mouse models express human APP under the Thy-1 promoter, thus glial cells from these mice are not suitable to test our hypothesis. To this aim, we are planning experiments with different AD mouse models -  not available yet - such asTg2576 or APP/PS1DE9 that express the APP-Swedish under the prion protein promoter.

Minor Points

While the authors discuss different aspects of FAD-linked APP Swedish mutation, a more detailed discussion on the type and location (i.e., domain) of the mutation, and how it might impact interactions at the molecular level would help capture the message.

 A graphical illustration should be sufficient, also depicting the event cascade through the Aβ accumulation. Similarly, a graphic depicting the role of ORAI2 and the impact on the levels of SOCE in AD is essential and should be added, perhaps in the discussion section.  

We thank the Reviewer for rising these interesting points. The Discussion has now been integrated accordingly and a graphical abstract and a new Figure 8 are now included. As far as the APP-Swedish mutation is concerned, we have now added a sentence in the Discussion (lines 365-368, pag 12, bottom) to highlight the fact that this mutation better mimics the sporadic forms of AD given the fact that it does not change the Ab42/Ab40 ratio but it simply increases the b-secretase activity. All the ratio changes, observed with SOCE modulators or ORAI2 down-regulation, can likely be linked to the changes occurring in the g-secretase microenvironment.

Reviewer 2 Report

Restoration of calcium homeostasis represents a valid alternative to targeting amyloid-β for development of novel Alzheimer’s therapies. The manuscript by Scremin et al described ORAI2 overexpression reduce SOCE, while downregulation increase SOCE and lead to reduction of Aβ42 in cultured neuroglioma cells.  It provides new insight of ORAI2 function and suggests a therapeutic potential. The manuscript can be improved by more concise and clear presentation. For example, in the abstract, the description γ-secretase can be moved to introduction, as the main results are on SOCE and ORAI.

Author Response

Dear Editor,

We thank you the Reviewers for the precious comments. Here are the detailed answers to the specific points. The revised version shows the relevant changes in yellow. Few typos have also been corrected.

Reviewer 2

Restoration of calcium homeostasis represents a valid alternative to targeting amyloid-β for development of novel Alzheimer’s therapies. The manuscript by Scremin et al described ORAI2 overexpression reduce SOCE, while downregulation increase SOCE and lead to reduction of Aβ42 in cultured neuroglioma cells.  It provides new insight of ORAI2 function and suggests a therapeutic potential. The manuscript can be improved by more concise and clear presentation. For example, in the abstract, the description γ-secretase can be moved to introduction, as the main results are on SOCE and ORAI.

We thank the Reviewer for the suggestions. The abstract has been modified accordingly, see lines 19-21. We carefully revised the manuscript, removing few unnecessary details (see line 164-165 and 183), and we hope that the new sentences, the graphical abstract and the novel figure 8 improve the readability, allowing to better catch the overall message.